# Bee Products as Interesting Natural Agents for the Prevention and Treatment of Common Cardiovascular Diseases

**DOI:** 10.3390/nu14112267

**Published:** 2022-05-28

**Authors:** Beata Olas

**Affiliations:** Department of General Biochemistry, Faculty of Biology and Environmental Protection, University of Lodz, Pomorska 141/3, 90-236 Lodz, Poland; beata.olas@biol.uni.lodz.pl

**Keywords:** bee products, cardiovascular diseases, hemostasis, oxidative stress, phenolic compounds

## Abstract

Apitherapy is a form of alternative therapy that relies on the use of bee products, i.e., honey, royal jelly, propolis, pollen, and bee venom (known as apitoxin), for the prevention and treatment of various diseases. Various in vitro and in vivo studies suggest that these products may be effective in the prophylaxis and treatment of cardiovascular diseases (CVDs). This mini-review of papers identified in various electronic databases describes new aspects of the bioactivity of certain bee products, viz. bee pollen, royal jelly, bee venom, propolis, and bee bread, as natural interesting products for the prevention and treatment of common CVDs.

## 1. Introduction

Apitherapy is an alternative therapy that relies on the use of bee products, such as honey, propolis, pollen, royal jelly, and bee venom (known as apitoxin), for the prevention and treatment of various diseases. Their use has a long and wide history, being described in Chinese, Egyptian, Russian, Korean, and Greek traditional medicine practices, dating back to the times of Galen and Hippocrates. Their use is also referred to in religious texts, such as the Bible and the Quran.

Humans have historically found a wide range of uses for bee products. For example, honey was used for food and religious offerings, propolis was used as an adhesive, pollen was used for plant breeding and other agricultural work, and beeswax was used in casting metals and making incendiary weapons [1]. More recently, these products have also been found to offer benefits in the prophylaxis and treatment of cardiovascular diseases (CVDs), such as atherosclerosis, hypertension, ischemia-reperfusion injury, and myocardial infarction [2,3,4,5,6]. This is a very important finding, as myocardial infarction is a major cause of death and disability worldwide and may often be the first manifestation of coronary artery disease. It may also occur repeatedly in patients with established disease [7,8].

The likelihood of CVDs is greatly increased by various exogenous and endogenous risk factors, such as blood platelet hyperactivation, hypercholesterolemia, oxidative stress, and obesity [9,10]; however, a number of dietary components, such as polyunsaturated fatty acids and phytosterols (for example, β-sitosterol), may play a protective role. In addition, phenolic compounds can also reduce the risk of CVDs. More specifically, consumption of phenolic compounds, especially flavonoids, is inversely associated with mortality associated with CVDs [11]. 

Various bee products may reduce the risk of CVDs through their antioxidant activity [6,12] and anti-platelet properties [6]. For example, honey possesses a range of cardioprotective properties thanks to its chemical content, particularly its phenolic compounds, including chrysin, quercetin, luteolin, and pinocembrin [6,13,14], which possess anti-platelet, antioxidant, and anti-inflammatory properties, among others. These have been observed both in vitro and in animal experiments, but only one study has been performed in vivo in a human model. This was carried out on healthy people. Meanwhile, the effects of honey and its key active components on CVDs are extensively characterized and described elsewhere [6]. In addition, it is unknown whether the pure phenolic compounds isolated from honey are more effective for the prophylaxis and treatment of CVDs than the honey itself. 

Some authors [15,16,17,18] describe mad honey having adverse effects on the cardiovascular system. For example, these effects include hypotension, bradycardia, and myocardial infarction. This mini-review describes new aspects of bioactivity of certain other bee products, viz. bee pollen, royal jelly, bee venom, propolis, and bee bread, and their potential use as natural products for the prevention and treatment of common cardiovascular diseases. The review was performed on the basis of a search of papers identified in the following electronic databases: PubMed, Scopus, ScienceDirect, Google Scholar, and Web of Knowledge. The last search was run on 30 March 2022. The following terms were used: “bee pollen”, “royal jelly”, “bee’s venom”, “propolis”, “bee bread”, “bee product and cardiovascular disease”.

## 2. Chemical Compounds Present in Bee Products and Their Cardioprotective Potential

### 2.1. Propolis

Propolis, a popular functional food originating from bees, is known to contain about 2000 compounds, but its chemical content depends on the origin and plant species. It comprises 45–55% phenolic compounds (phenolic acid and its esters, flavonoids), 10% essential oils, 25–35% waxes and fatty acids, 5% pollen containing amino acids, and 5% various other substances (sugars, vitamins, and minerals) [19,20].

Majiene et al. [21] noted the inhibitory effect of propolis water solution on mitochondrial respiration. 

Zhang et al. [22] found the water extract of propolis (25–300 mg/L) to inhibit human blood platelet aggregation in vitro against the following platelet agonists: 10 µM ADP, 5 µg/mL collagen, and 50 µM thrombin receptor activator peptide (TRAP). Platelet aggregation was measured in platelet-rich plasma by turbidimetry. 

Bojic et al. [20] report that propolis ethanol extract (total phenolic compound content: 136.14 mg/g; total flavonoid content: 19.28 mg/g) also demonstrated anti-aggregatory properties in vitro at concentrations of 5 µM to 10.4 mM. Platelet aggregation was induced by ADP in whole-blood samples of ten healthy volunteers, and this process was analyzed by impendence analyzer. Bojic et al. [20,23], Hubbard et al. [24], and Faggio et al. [25] propose that flavonoids, being the major constituents of propolis, are responsible for its anti-platelet properties. They are known to prevent thrombin-, ADP-, and collagen-induced blood platelet aggregation by inhibiting various enzymes within intracellular signaling pathways, including phospholipase C and A_2_, and cyclooxygenase. In addition, flavonoids may reduce the oxidative burst and decrease nitric oxide generation. 

Interestingly, various authors indicate that propolis has a cardioprotective potential in vivo [19,25]. Results of He et al. [26] indicate the protective action of total flavonoids of propolis (25 and 50 mg/kg/day) on pathological cardiac hypertrophy and heart failure in mice. The protective mechanism of the tested propolis included the phosphoinositide 3-kinase (PI3-K)/AKT signaling pathway. Recently, Chao et al. [27] reported the effects of propolis on neointimal formation in a rabbit carotid artery balloon-induced injury model with hypercholesterolemia. The rabbits (n = 24) were fed a 1% high-cholesterol diet for six weeks after surgery, and propolis (125 mg/kg and 250 mg/kg) was given orally for six weeks. At the two used doses, propolis was found to reduce body weight and blood lipid level in the rabbits. Propolis also inhibited neointimal hyperplasia and carotid stenosis in these animals and attenuated lipid deposition, collagen fiber, and reduced oxidative stress. The effect of propolis on oxidative stress was determined by various biomarkers, including the activity of superoxide dismutase and nitric oxide in plasma. Interestingly, propolis was found to inhibit the expression of Toll-like receptor 4 (TLR4), nuclear factor-κB (NF-κB), and tumor necrosis factor-α (TNF-α), and authors suggest that propolis could improve the level of vascular inflammation through the TLR4/NFκB pathway. The results of this study suggest that propolis may form the basis of future studies aimed at preventing and treating vascular restenosis.

Wang et al. [28] observed that total flavonoids of propolis have inhibitory action on apoptosis of myocardial cells of chronic heart failure stimulated by adriamycin in rats. Moreover, using a propolis preparation may regulate Cx43 expression, especially the phosphorylation status.

One of the most active components of propolis is caffeic acid phenethyl ester (CAPE; C_17_H_16_O_4_; (E)—3-(3,4-dihydroxyphenyl)-2-2-propionic acid, 2-phenylethyl 3-(3,4-dihydroxyphenyl)-2-propenoate). CAPE was first identified as a component of propolis and can be extracted from propolis by various methods. A review by Tolba et al. [19] described the broad spectrum of pharmacological properties of this compound. For example, CAPE not only appeared to possess antioxidant activity but demonstrated protective effects against ischemia-reperfusion— an injury induced in various tissues, including heart, brain, liver, and colon.

### 2.2. Bee Pollen

Bee pollen is obtained from the agglutination of flower pollens with the nectar and salivary substances of honeybees. Bee pollen has high nutritious value and is believed to contain over 250 substances; however, their type and function depend on the plant species from which the pollen was collected, as well as the climate zone and the season. Dandelion and black mustard pollens are the richest in lipids (>10%). On the other hand, insignificant amounts of these substances are found in corn pollen and hazelnut pollen [29]. The key role in the biological activity of bee pollen is determined by its phenolic compound content, typically comprising 1.6% of the total mass, mainly phenolic acids (0.2%) and flavonoids (1.4%) [30,31]. The main contents of bee pollen and other bee products are presented in Figure 1. For example, some results [32,33,34] indicate that royal jelly proteins have antihypertensive action and hypocholesterolemic properties.

The bee pollen of *Schisandra chinensis* (Turcz.) is often used as a functional food in China. Zhang et al. [36,37] found this pollen to have the strongest total antioxidant capacity among ten tested types of pollen. Shi et al. [31] also studied the chemical profile of *S. chinensis* bee pollen extract and its effect on H_2_O_2_-induced apoptosis in H9c2 cardiomyocytes. They identified two carbohydrates, three nucleotides, and nine quinic-acid-containing derivatives. The tested extract (12.5; 25 and 50 µg/mL) was also found to offer favorable antioxidant properties, such as increasing myocardial superoxide dismutase (SOD) activity and glutathione (GSH) concentration and decreasing myocardial malondialdehyde (MDA) level in vitro. 

In in vivo studies, Shen et al. [8] studied the antioxidant and cardioprotective actions of *S. chinensis* bee pollen extract on acute myocardial infarction stimulated by isoprenaline in rats. The authors propose that the cardioprotective action of the tested extract may be related to its antioxidative properties; they also suggest that up-regulation of antioxidant enzyme activity in rat heart, under the control of the transcription factor Nrf2, may be an adaptive mechanism to overcome isoprenaline-induced oxidative stress and myocardial infarction. In addition, rats treated with the used extract demonstrated higher protein expression of heme oxygenase-1 and Bcl2 in the heart but lower protein expression of Bax. These results may indicate that bee pollen extract could reduce cardiocyte apoptosis through the regulation of Bcl-2 and Bax. Other authors have also reported the antioxidant potential of bee pollen [27,36,37].

Rzepecka-Stojko et al. [30] propose that the polyphenol-rich extract from bee pollen may have anti-atherogenic properties. The extract was given in two doses: 0.1 and 1 g/kg body mass (BM). Atherosclerosis was induced by a high-fat diet in Apo-knockout mice. The studies were conducted over a period of 16 weeks. It was found that at both tested amounts, the polyphenol-rich extract modulated the lipid profile in the tested animals by lowering the total cholesterol level. It also reduced oxidative stress by various mechanisms, including lowering levels of oxidized low-density lipoprotein (LDL), and appeared to protect the coronary arteries by limiting the development of atherosclerosis (at 0.1 g/kg BM) and completely preventing its occurrence (at 1 g/kg BM). The authors also propose that bee pollen extract inhibits the development of atherosclerosis in the tested animals by reducing the activity of angiotensin-converting enzyme (ACE) and decreasing the level of angiotensin II.

Kasianenko et al. [38] studied the efficacy of various bee products (honey, pollen, and bee bread) as treatment for patients with atherogenic dyslipidemia. This experiment included 157 patients (64 men and 93 women). It is important to note that none of the participants receiving bee food products demonstrated any allergy or individual resistance to pollen, bee bread, or honey. Significant hypolipidemic activity was observed for the two tested groups, viz. the patients receiving pollen and honey (total cholesterol decreased by 18.3%) and those receiving bee bread (total cholesterol decreased by 15.7%). In addition, it was found that among overweight (BMI: 25–30) and obese (BMI > 30) patients, pollen and honey treatment only improved blood lipid composition in cases where body mass was lost.

Gulhan et al. [39] studied the therapeutic and protective effects of ethanolic extracts of pollen and propolis on the reproductive function in (L-N^G^-nitro arginine methyl ester (L-NAME)-induced hypertensive male rats. The rats were separated into four groups: (group 1) control, (group 2) L-NAME, (group 3) L-NAME and propolis, and (group 4) L-NAME and pollen. It was found that the tested bee products appeared to influence reproductive function by reducing the activity of the inflammatory pathways leading to hypertension.

### 2.3. Royal Jelly

Royal jelly is a well-known functional food. In the hive, it is produced by nurse bees to feed young worker larvae and queen bees. It contains water (60–70% *w*/*w*), sugars (7–18% *w*/*w*), lipids (3–8% *w*/*w*), and proteins (9–18% *w*/*w*); it can be seen that protein constitutes about 50% of its dry weight. About 80% of this protein content consists of major royal jelly proteins (MRJPs) 1–9, with MRJP1 being the most abundant, making up approximately 48% of the water-soluble proteins in royal jelly [40]. The jelly also contains a number of amino acids, minerals, hormones, vitamins, and phenolic compounds (total content: 23.3 µg/mg), including flavonoids (total content: 1.28 µg/mg) [41]. Phytosterols, such campesterol (67%), β-sitosterol (19–24%), isofucosterol (9–16%), and desmosterol (0.5–4.5%), have also been found in royal jelly [42].

Royal jelly is produced for commercial purposes, with a significantly higher market value than honey and pollen. Its largest global producer and exporter is China. Studies indicate that royal jelly has a wide range of biological properties, including antioxidant and anti-hypertensive potential. More details about its chemical components and its biological properties are reviewed by Ahmad et al. [35]. A few experimental papers have examined its cardioprotective properties, including its anti-hypercholesterolemic and anti-hypertension activity [32,33,43,44,45,46].

Fan et al. [40] propose that major royal jelly protein 1 (MRJP1) may be used to control hypertension. Its mechanism of action may include regulating the activity of vascular smooth muscle cells (VSMCs), a key component of the arterial well that plays an important role in blood pressure regulation, as well as contraction, migration, and proliferation.

Royal jelly also appears to have hypocholesterolemic properties. Kashima et al. [34] identified the hypocholesterolemic protein MRJP1 in the royal jelly, which exhibits greater hypocholesterolemic activity than β-sitosterol. It was found that 600 mg/kg/day dietary MRJP1 administered for seven days influenced serum cholesterol concentration in rats. Other studies have found that intervention with royal jelly (350 mg/capsule/day, for three months) lowers serum total cholesterol and LDL levels in healthy, mildly hypercholesterolemic adults. The authors suggest that the tested preparation may regulate the blood lipid profile by ameliorating the level of dehydroepiandrosterone sulphate [47]. 

Lambrinoudaki et al. [48] studied the effect of royal jelly (150 mg/day) on cardiovascular and bone turnover markers in clinically healthy postmenopausal women (n = 36). The participants received royal jelly for three months. The royal jelly used in the study was found to be rich in medium-chain fatty acids, i.e., compounds with hypolipidemic properties, comprising 63% of the dry weight fatty content. The treatment resulted in a significant increase in HDL cholesterol and a significant decrease in LDL and total cholesterol. The authors also suggest that royal jelly supplementation may offer an alternative method of managing menopause-associated dyslipidemia.

A recent meta-analysis by Hadi et al. [49] also suggested that royal jelly consumption may improve lipid parameters. A review of PUBMED, the Cochrane Library, Scopus, Web of Science, and Google Scholar was performed to identify clinical trials investigating the efficacy of royal jelly on adult blood lipid parameters until July 2017. The resulting pooled analysis of six trials found that while royal jelly reduces total cholesterol blood levels, no significant change was observed in triglyceride, LDL cholesterol, or HDL cholesterol blood concentrations. 

Recently, the results of Aslan et al. [50] have also demonstrated that royal jelly has cardioprotective potential against heart tissue damage induced by fluoride (by activating the nrf-2/NF-κB and bcl-2/bax signaling pathway). This experiment included 42 rats. Royal jelly was given in one dose: 100 mg/kg five times a week for 8 weeks. 

Georgiev et al. [51] examined the effect of a food supplement named Melbrosia, containing a combination of flower pollen, perga (fermented flower pollen), and royal jelly in 55 postmenopausal women with menopausal complaints over three months. For the first two weeks, the patients were treated with two capsules of Melbrosia once a day and then one capsule daily for ten days. The treatment was found to result in a significant decrease in total cholesterol (TC) and LDL cholesterol and a significant increase in HDL cholesterol, which may have an influence on CVD risk. 

### 2.4. Bee Venom

Bee venom can be introduced into the human body by direct bee stings and by manual injection. It contains different active compounds, including peptides (melittin, apamin, adolapin), enzymes (for example, phospholipase A_2_), amines (for example, epinephrine and histamine), and minerals. The composition of the venom is 88% water and only 0.1 µg dry venom [5,52,53]. More details about the chemical content of bee venom, its biological properties, and mechanisms of action are given in a review by Wehbe et al. [5]. However, no information has been given about the role of bee venom in CVDs, but the results of Yook et al. [54] show that sweet bee venom and bee venom may modulate heart rate variability. Guimaraes et al. [55] also observed the changes in mean arterial pressure and heart rate in rats inoculated with Africanized bee venom.

Wang et al. [56] indicate that melittin, a major polypeptide in bee venom, ameliorates coxsackievirus B3 (CVB3)-induced myocarditis, which is characterized by myocardial inflammation. The disease causes sudden death, especially in children and young people. BALB/c mice were injected with CVB3 to develop viral myocarditis and were then treated with intraperitoneal melittin (0.1 mg/kg) for seven days. It was found that melittin treatment reduced myocardial cell apoptosis, decreased the expression of bax and caspase-3, and increased that of bcl-2. It also improved cardiac function, as indicated by echocardiography. The authors suggest that its mechanism of action may be associated with activation of the histone deacetylase 2 (HDAC2)-mediated GSH-3β/nuclear factor E2 (Nrf2)/antioxidant response element (ARE) signaling pathway. 

### 2.5. Bee Bread

Nagai et al. [57] indicate that hydrolysates from bee bread have antioxidant properties (similar to 1 mM α-tocopherol) measured by scavenging the superoxide anion radicals and hydroxyl radicals. The bread was also found to inhibit angiotensin I-converting enzyme (ACE) in a similar way to other fermented foods, such as miso, sake, natto, and fish sauce.

## 3. Conclusions

Both honey and other bee products appear to have some cardioprotective potential. The biological properties of selected bee products are given in Figure 2, while the cardioprotective potential of propolis, bee pollen, and royal jelly are summarized in Table 1. Unfortunately, while this potential has been observed both in vitro and in animal and human models, only a few in vivo studies have been performed in a human model. In addition, there is as yet no concrete clinical evidence for the efficacy or safety of apitherapy treatments, nor any recommendations regarding the effective or safe doses of bee products for the prophylaxis and treatment of CVDs. Furthermore, the reviewed studies employed different types of bee products, some of which were of unknown composition. 

Despite this, this review sheds new light on the cardioprotective mechanisms of compounds present in bee products. While these mechanisms are complex and generally remain unknown, but phenolic compounds in particular may play an important role in these mechanisms (Figure 2). For example, the phenolic components of propolis may decrease reactive oxygen species (ROS) and nitric oxide (NO) production and reduce the activity of cyclooxygenase, phospholipase A_2_ and C. Moreover, phenolic compounds of propolis may reduce the expression of TLR4 and NF-κB. However, further studies are needed to clarity the mechanisms of their action.

Even so, it is important to note that while no adverse effects were observed in any of the subjects treated with bee products, they have the potential to induce allergic reactions. Such reactions can take place in the cardiovascular system and may lead to myocardial ischemia.

## Figures and Tables

**Figure 1 nutrients-14-02267-f001:**
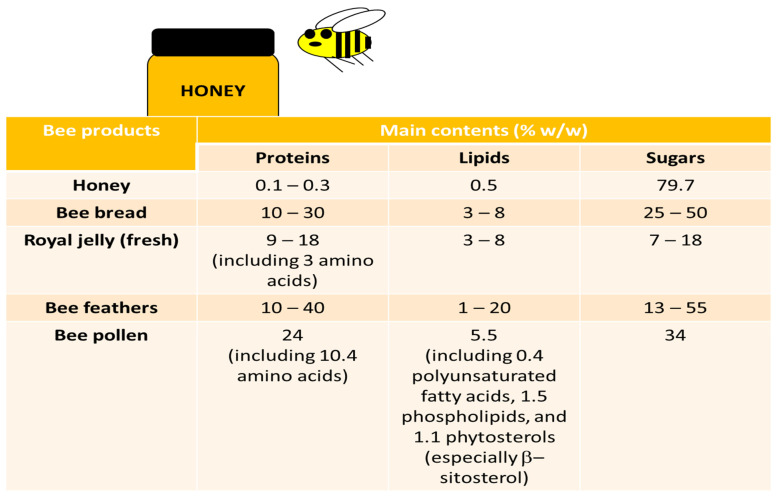
Main contents of selected bee products (Adapted from ref. [35]).

**Figure 2 nutrients-14-02267-f002:**
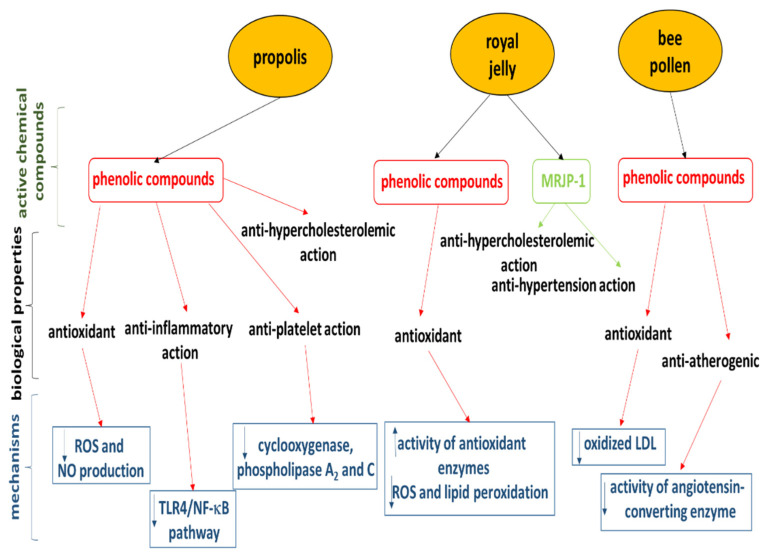
A schematic representation of the biological substances in selected bee products and their cardioprotective mechanisms.

**Table 1 nutrients-14-02267-t001:** Cardioprotective potential of bee products in various in vitro and in vivo models.

Bee Product	Investigated Roles	References
Propolis		
Water extract (CAPE was the highest in this extract, followed by galangin, ferulic acid, quercetin, kaempferol, and apigenin): 25 and 300 mg/L	Anti-aggregatory potential (in vitro; three healthy volunteers)	[22]
Ethanol extract (content of total phenolic compounds: 136.14 mg/g, and content of total flavonoids: 19.28 mg/g): 5–10.4 mM	Anti-aggregatory potential (in vitro; ten healthy volunteers)	[20]
Propolis (3-O-acetyl pinobanksin, chrysin, pinocembrin, pinobanksin, and CAPE—the five most abundant components): 125 and 250 mg/kg/day	Reducing body weight and the level of blood lipids (in vivo; hypercholesterolemic rabbits (n = 24))	[26]
Total flavonoids of propolis: 25 and 50 mg/kg/day	Attenuating adverse cardiac dysfunction and hypertrophy (in vivo, mice)	[27]
Total flavonoids of propolis	Inhibitory action on apoptosis of myocardial cells of chronic heart failure (in vivo; rats (n = 6))	[28]
Proplis water solution (total phenolics: 188 mg/mL): 9, 33, 63 and 125 µg/mL	Inhibitory effect on mitochondrial respiration (in vitro; heart mitochondria)	[21]
Bee pollen		
Bee pollen of *S. chinensis* (Turcz.) extract (two carbohydrates, three nucleotides, and nine quinic-acid-containing derivatives were identified): 12.5, 25 and 50 µg/mL	Antioxidant effect (in vitro; H9c2 cardiomyocytes)	[31]
Bee pollen of *S. chinensis* (Turcz.) extract (one major compound was identified as uridine): 270, 600, 1200 and 1800 mg/kg/day	Antioxidant and cardioprotective effect (in vivo, rats with myocardial infarction induced by isoprenaline, (n = 36))	[8]
Polyphenol-rich extract from bee pollen (chemical content: undefined): 0.1 and 1 g/kg BM	Antioxidant and anti-atherogenic effect (in vivo, Apo-knockout mice with atherosclerosis induced by a high-fat diet, (n = 60))	[30]
Royal jelly		
Major royal jelly protein 1 (chemical content: undefined)	Anti-hypertension effect (in vitro, mouse vascular muscle cells (n = 3))	[40]
Royal jelly (chemical content: the total protein—142.8 ± 0.35 mg/g; MRJP1 and 2—two major proteins), used doses: 350 mg/capsule, 3 months	Anti-hypercholesterolemic effect (in vivo, healthy mildly hypercholesterolemic adults (n = 40))	[47]
Royal jelly, used doses: 150 mg/day for three months	Anti-hypercholesterolemic effect (in vivo, postmenopausal healthy women (n = 36))	[48]
Royal jelly, used doses: 100 mg/kg five times a week for 8 weeks	Antioxidant action (in vivo, rats (n = 42))	[50]
MRJP1from royal jelly, used doses: 600 mg/kg/day, for 7 days	Anti-hypercholesterolemic effect (in vivo, rats (n = 10))	[34]
Different bee products (together)		
Honey, pollen, and bee bread (chemical content and used doses: undefined)	Hypolipidemic effect for two tested groups: (1) patients taking bee pollen and honey; (2) patients taking bee bread (in vivo, patients with atherogenic dyslipidemia (n = 157))	[38]
Bee pollen and propolis as ethanolic extracts (chemical content and used doses: undefined)	Anti-hypertension effect (in vivo, rats with hypertension induced by L-NAME (n = 28))	[39]

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
