# Peer review of "Bee Products as Interesting Natural Agents for the Prevention and Treatment of Common Cardiovascular Diseases"

_nutrients, 2022, doi:10.3390/nu14112267_

Round 1

Reviewer 1 Report

In this manuscript, the authors reviewed papers in various electronic databases about biological activities of bee products related to the prevention and treatment of cardiovascular diseases (CVDs). The topic is interesting since bee products can be incorporated into the diet or used as pharmacological agents as an alternative therapy for CVDs. Similar reviews on biological activities of bee products have been published elsewhere but none of them is focused on the protective roles of bee products in CDVs, therefore this review identified a gap in knowledge and is still of interest. However, the manuscript has some major flaws:

1.      In the Abstract, the aim of this review paper is not clearly stated. The question is whether honey is within the scope of this review or not. If this review focuses on bee products other than honey, then figure 1 should be deleted.

2.      In the Introduction, the author states that this review aims to describe new aspects of bioactivity of certain bee products, however, active ingredients in these products and their main biological activities related to CDVs is not summarized. The author identify the key components of bee products for protection of CVDs are phenolic compounds, are phenolic compounds different in propolis, been pollen, or royal jelly?

3.      In the Introduction, the third paragraph, the last sentence, the reference of De Gaetano et al. is not appropriate because this paper is mainly about beer consumption and CVD risk.

4.      Figure 2, are these main nutrients listed in different bee products shown to be beneficial to CVDs? Are they worth further investigation in this aspect? Please comment.

5.      In the section of 2.2. Bee pollen, the third paragraph, line 3-10, too much experimental detail is listed, it is not necessary. The author only needs to summarize the main findings and conclusions. The same is true for the forth and the fifth paragraphs in 2.2.

6.      The majority reference in 2.3. Royal jelly section is not within the last 5 years.  If the comprehensive biological activities of royal jelly are already reviewed by Ahmad et al., bioactivities related to CDVs should be only briefly summarized.

7.      Figure 3 and Figure 4 is overlapping, figure 3 may be deleted.

8.      In Table 1, the listed reference papers in Royal jelly part are too old, only one within the last 5 years.

9.      Figure 4 is not easy to follow and should be restructured to be more visually appealing. Some contents are redundant and confusing, for example, reduced production of ROS and NO production is part of the antioxidant activity which can be listed as one mechanism for cardio-protection. Anti-inflammatory activity is also an important mechanism for CVD protection by propolis, but it is not listed in figure 4. The figure should be restructured in a way that is easy to understand. Three layers can be included: active chemical components, biological effects for CDV protection, and mechanisms for bioactivity.

10.  In the Introduction, the author stated “This mini-review describes new aspects of the bioactivity of certain bee products, vis. bee feathers…”, but there is no content about bee feathers in the text except in figure 2 and nothing about its biological activity.

11.  There are no clear statements about the cardio-protective role of bee products and future research needs in conclusion section.

12.  In most parts of the manuscript, the author just listed experimental details of relevant papers. There are no coherent statements and in-depth analysis of involved mechanisms.

Minor problems:

13.  In the Introduction, third paragraph, line 4, “omega” is never used alone, the commonly used term is “omega-3”, “omega-6”or “PUFA”.

14.  In 2.2., the last line, please describe in detail the plant source of these bee pollen oil. 

Author Response

In this manuscript, the authors reviewed papers in various electronic databases about biological activities of bee products related to the prevention and treatment of cardiovascular diseases (CVDs). The topic is interesting since bee products can be incorporated into the diet or used as pharmacological agents as an alternative therapy for CVDs. Similar reviews on biological activities of bee products have been published elsewhere but none of them is focused on the protective roles of bee products in CDVs, therefore this review identified a gap in knowledge and is still of interest. However, the manuscript has some major flaws:

Response: I thank the Reviewer for helpful comments. Moreover, I agree with the comment of Reviewer, and this wrong statement was corrected.

  1. In the Abstract, the aim of this review paper is not clearly stated. The question is whether honey is within the scope of this review or not. If this review focuses on bee products other than honey, then figure 1 should be deleted.

Response: I have changed the Abstract and deleted Fig. 1 (honey).

  1. In the Introduction, the author states that this review aims to describe new aspects of bioactivity of certain bee products, however, active ingredients in these products and their main biological activities related to CDVs is not summarized. The author identify the key components of bee products for protection of CVDs are phenolic compounds, are phenolic compounds different in propolis, been pollen, or royal jelly?

Response: There is more information about chemical content (including phenolic compounds) of different bee products in the next chapter: “Chemical compounds….”

  1. In the Introduction, the third paragraph, the last sentence, the reference of De Gaetano et al. is not appropriate because this paper is mainly about beer consumption and CVD risk.

Response: I have deleted this paper.

  1. Figure 2, are these main nutrients listed in different bee products shown to be beneficial to CVDs? Are they worth further investigation in this aspect? Please comment.

Response: I have added short information about it: “For example, some results (Matsui et al., 2002; Tokunaga et al., 2004; Kashima et al., 2014) indicate that royal jelly proteins have antihypertenisive action and hypocholesterolaemic properties.”

  1. In the section of 2.2. Bee pollen, the third paragraph, line 3-10, too much experimental detail is listed, it is not necessary. The author only needs to summarize the main findings and conclusions. The same is true for the forth and the fifth paragraphs in 2.2.

Response: I have deleted experimental details.

  1. The majority reference in 2.3. Royal jelly section is not within the last 5 years. If the comprehensive biological activities of royal jelly are already reviewed by Ahmad et al., bioactivities related to CDVs should be only briefly summarized.

Response: I have corrected the chapter 2.3. I have described results of papers (the last 5 years).

  1. Figure 3 and Figure 4 is overlapping, figure 3 may be deleted.

Response: I have deleted figure 3.

  1. In Table 1, the listed reference papers in Royal jelly part are too old, only one within the last 5 years.

Response: I have deleted papers (1995, 2002, 2004, 2007).

  1. Figure 4 is not easy to follow and should be restructured to be more visually appealing. Some contents are redundant and confusing, for example, reduced production of ROS and NO production is part of the antioxidant activity which can be listed as one mechanism for cardio-protection. Anti-inflammatory activity is also an important mechanism for CVD protection by propolis, but it is not listed in figure 4. The figure should be restructured in a way that is easy to understand. Three layers can be included: active chemical components, biological effects for CDV protection, and mechanisms for bioactivity.

Response: I have modified this figure (now, it is Fig. 2) and added new information.

  1. In the Introduction, the author stated “This mini-review describes new aspects of the bioactivity of certain bee products, vis. bee feathers…”, but there is no content about bee feathers in the text except in figure 2 and nothing about its biological activity.

Response: I have deleted “bee feathers”, because there is no information about its effects of cardiovascular system and diseases.

  1. There are no clear statements about the cardio-protective role of bee products and future research needs in conclusion section.

Response: I have modified conclusion section.

  1. In most parts of the manuscript, the author just listed experimental details of relevant papers. There are no coherent statements and in-depth analysis of involved mechanisms.

Response: I have added new information about it in conclusion section.

Minor problems:

  1. In the Introduction, third paragraph, line 4, “omega” is never used alone, the commonly used term is “omega-3”, “omega-6”or “PUFA”.

Response: I have deleted “omega”, and used “polyunsaturated fatty acids”.

  1. In 2.2., the last line, please describe in detail the plant source of these bee pollen oil.

Response: I have added more details about it: “Dandelion and black mustard pollens are the richest in lipids (>10%). On the other hand, insignificant amounts of these substances are found in corn pollen and hazelnut pollen (Kedzia, 2008).”

Reviewer 2 Report

In this review, the authors aimed to provide literature information on bee products as interesting natural agents for the prevention and treatment of common cardiovascular diseases.

The subject of the manuscript is partly original and falls within the scope of the journal. Moreover, I think this review could provide valuable data for scientific literature and the readers of the journal. However, there is more research in the literature on the subject, and the articles from other databases such as “google academic” should be included in the review. 

  1. There are many articles related to bee and cardiovascular diseases in the literature. Some of them are given below. These articles should be considered in order to increase the scientific value of the review.

Hussain, N. H. N., Sulaiman, S. A., Hassan, I. I., Kadir, A. A., Nor, N. M., Ismail, S. B., ... & Musa, K. I. (2012). Randomized controlled trial on the effects of tualang honey and hormonal replacement therapy (HRT) on cardiovascular risk factors, hormonal profiles and bone density among postmenopausal women: a pilot study. Journal of Food Research, 1(2), 171.

Marsh, N. A., & Whaler, B. C. (1980). The effects of honey bee (Apis mellifera L.) venom and two of its constituents, melittin and phospholipase A2, on the cardiovascular system of the rat. Toxicon, 18(4), 427-435.

Martina, S. J., Ramar, L. A., Silaban, M. R., Luthfi, M., & Govindan, P. A. (2019). Antiplatelet effectivity between aspirin with honey on cardiovascular disease based on bleeding time taken on mice. Open Access Macedonian Journal of Medical Sciences, 7(20), 3416.

Asaduzzaman, M., Sohanur Rahman, M., Munira, S., Muedur Rahman, M., Hasan, M., Siddique, M. A. H., ... & Islam, A. M. (2015). Effects of honey supplementation on hepatic and cardiovascular disease (CVD) marker in streptozotocin-induced diabetic rats. J Diabetes Metab, 6(592), 2.

Guimarães, J. V., Costa, R. S., Machado, B. H., & Reis, M. A. D. (2004). Cardiovascular profile after intravenous injection of Africanized bee venom in awake rats. Revista do Instituto de Medicina Tropical de São Paulo, 46, 55-58.

Majiene, D., Trumbeckaite, S., Savickas, A., & Toleikis, A. (2006). Influence of propolis water solution on heart mitochondrial function. Journal of pharmacy and pharmacology, 58(5), 709-713.

Wang, H. H., Zeng, J., Wang, H. Z., Jiang, Y. X., Wang, J., & Zhou, P. P. (2015). Effects of total flavonoids of propolis on apoptosis of myocardial cells of chronic heart failure and its possible mechanism in rats. Zhongguo Ying Yong Sheng li xue za zhi= Zhongguo Yingyong Shenglixue Zazhi= Chinese Journal of Applied Physiology, 31(3), 201-206.

He, T., Sui, X., Sun, W., Yang, P., Sui, D., Cui, H., ... & Sun, G. (2018). GW29-e0190 The Protective Effects and Mechanism of Total Flavonoids of Propolis on Pathological Cardiac Hypertrophy and Heart Failure in Mice. Journal of the American College of Cardiology, 72(16S), C161-C161.

Lim, O. Z., Yeoh, B. S., Omar, N., Mohamed, M., Zin, A. A. M., & Ahmad, R. (2020). Synergistic Cardioprotective Activity of Stingless Bee Propolis and Metformin Through Modulation of Anti-Oxidants in Diabetic Heart: The Relationship Between Anti-Oxidants and Oxidative Stress. The Malaysian Journal of Medical Sciences, 27, 3-4.

Alagwu, E. A., Okwara, J. E., Nneli, R. O., & Osim, E. E. (2014). Effect of honey intake on serum cholesterol, triglycerides and lipoprotein levels in albino rats and potential benefits on risks of coronary heart disease.

Najafi, M., Shaseb, E., Ghaffary, S., Fakhrju, A., & Eteraf Oskouei, T. (2011). Effects of chronic oral administration of natural honey on ischemia/reperfusion-induced arrhythmias in isolated rat heart. Iranian Journal of Basic Medical Sciences, 14(1), 75-81.

Aluko, E. O., Olubobokun, T. H., Enobong, I. B., & Atang, D. E. (2013). Comparative study of effect of honey on blood pressure and heart rate in healthy male and female subjects. British Journal of Medicine and Medical Research, 3(4), 2214.

Najafi, M., Zahednezhad, F., Samadzadeh, M., & Vaez, H. (2012). Zero flow global ischemia-induced injuries in rat heart are attenuated by natural honey. Advanced Pharmaceutical Bulletin, 2(2), 165.

Ebrahimi, M., Dehghani, F., Farhadian, N., Karimi, M., & Golmohammadzadeh, S. (2017). Investigating the anti-apoptotic effect of sesame oil and honey in a novel nanostructure form for treatment of heart failure. Nanomedicine Journal, 4(4), 245-253.

Aslan, A., Beyaz, S., Gok, O., Can, M. I., Parlak, G., Ozercan, I. H., & Gundogdu, R. (2021). Royal jelly abrogates flouride-induced oxidative damage in rat heart tissue by activating of the nrf-2/NF-κB and bcl-2/bax pathway. Toxicology Mechanisms and Methods, 31(9), 644-654.

Yook, T. H., Yu, J. S., & Jung, H. S. (2008). Effects of sweet bee venom and bee venom on the heart rate variability. Journal of Pharmacopuncture, 11(1), 41-54.

  1. P2: “This was carried out on healthy and young and people”. Should be revised.
  2. P4: Figure 2. I think it is a table, not a figure.
  3. P8, Figure 3: This figure does not make any sense and should be removed.
  4. Some types of honey (mad-honey) have adverse effects on the cardiovascular system. Brief information could be given in the article considering the articles given below.

SOHN, C. H., Kim, W., Ahn, S., OH, B. J., KIM, W. Y., & LIM, K. S. (2005). Three cases of mad-honey poisoning presenting with cardiovascular emergencies. Journal of the Korean Society of Emergency Medicine, 322-325.

Shrestha, T. M., Nepal, G., Shing, Y. K., & Shrestha, L. (2018). Cardiovascular, psychiatric, and neurological phenomena seen in mad honey disease: a clinical case report. Clinical Case Reports, 6(12), 2355.

Ko, Y. G., Kim, K. H., Kim, A. J., Shin, D. W., Park, J. S., Roh, J. Y., & Ahn, J. Y. (2006). Two Cases of Mad-Honey Poisoning with Cardiovascular Symptom. Journal of The Korean Society of Clinical Toxicology, 4(1), 78-81.

Setareh-Shenas, S., Kaplin, S., Bania, T. C., & Kornberg, R. (2019). A Rare Case of Mad Honey Disease: A Reversible Cause of Complete Heart Block. JACC: Case Reports, 1(4), 579-582.

Author Response

In this review, the authors aimed to provide literature information on bee products as interesting natural agents for the prevention and treatment of common cardiovascular diseases.

The subject of the manuscript is partly original and falls within the scope of the journal. Moreover, I think this review could provide valuable data for scientific literature and the readers of the journal. However, there is more research in the literature on the subject, and the articles from other databases such as “google academic” should be included in the review. 

Response: I thank the Reviewer for helpful comments. Moreover, I agree with the comment of Reviewer, and this wrong statement was corrected. I have also citied the articles from Google Scholar.

  1. There are many articles related to bee and cardiovascular diseases in the literature. Some of them are given below. These articles should be considered in order to increase the scientific value of the review.

Hussain, N. H. N., Sulaiman, S. A., Hassan, I. I., Kadir, A. A., Nor, N. M., Ismail, S. B., ... & Musa, K. I. (2012). Randomized controlled trial on the effects of tualang honey and hormonal replacement therapy (HRT) on cardiovascular risk factors, hormonal profiles and bone density among postmenopausal women: a pilot study. Journal of Food Research, 1(2), 171.

Marsh, N. A., & Whaler, B. C. (1980). The effects of honey bee (Apis mellifera L.) venom and two of its constituents, melittin and phospholipase A2, on the cardiovascular system of the rat. Toxicon, 18(4), 427-435.

Martina, S. J., Ramar, L. A., Silaban, M. R., Luthfi, M., & Govindan, P. A. (2019). Antiplatelet effectivity between aspirin with honey on cardiovascular disease based on bleeding time taken on mice. Open Access Macedonian Journal of Medical Sciences, 7(20), 3416.

Asaduzzaman, M., Sohanur Rahman, M., Munira, S., Muedur Rahman, M., Hasan, M., Siddique, M. A. H., ... & Islam, A. M. (2015). Effects of honey supplementation on hepatic and cardiovascular disease (CVD) marker in streptozotocin-induced diabetic rats. J Diabetes Metab, 6(592), 2.

Guimarães, J. V., Costa, R. S., Machado, B. H., & Reis, M. A. D. (2004). Cardiovascular profile after intravenous injection of Africanized bee venom in awake rats. Revista do Instituto de Medicina Tropical de São Paulo, 46, 55-58.

Majiene, D., Trumbeckaite, S., Savickas, A., & Toleikis, A. (2006). Influence of propolis water solution on heart mitochondrial function. Journal of pharmacy and pharmacology, 58(5), 709-713.

Wang, H. H., Zeng, J., Wang, H. Z., Jiang, Y. X., Wang, J., & Zhou, P. P. (2015). Effects of total flavonoids of propolis on apoptosis of myocardial cells of chronic heart failure and its possible mechanism in rats. Zhongguo Ying Yong Sheng li xue za zhi= Zhongguo Yingyong Shenglixue Zazhi= Chinese Journal of Applied Physiology, 31(3), 201-206.

He, T., Sui, X., Sun, W., Yang, P., Sui, D., Cui, H., ... & Sun, G. (2018). GW29-e0190 The Protective Effects and Mechanism of Total Flavonoids of Propolis on Pathological Cardiac Hypertrophy and Heart Failure in Mice. Journal of the American College of Cardiology, 72(16S), C161-C161.

Lim, O. Z., Yeoh, B. S., Omar, N., Mohamed, M., Zin, A. A. M., & Ahmad, R. (2020). Synergistic Cardioprotective Activity of Stingless Bee Propolis and Metformin Through Modulation of Anti-Oxidants in Diabetic Heart: The Relationship Between Anti-Oxidants and Oxidative Stress. The Malaysian Journal of Medical Sciences, 27, 3-4.

Alagwu, E. A., Okwara, J. E., Nneli, R. O., & Osim, E. E. (2014). Effect of honey intake on serum cholesterol, triglycerides and lipoprotein levels in albino rats and potential benefits on risks of coronary heart disease.

Najafi, M., Shaseb, E., Ghaffary, S., Fakhrju, A., & Eteraf Oskouei, T. (2011). Effects of chronic oral administration of natural honey on ischemia/reperfusion-induced arrhythmias in isolated rat heart. Iranian Journal of Basic Medical Sciences, 14(1), 75-81.

Aluko, E. O., Olubobokun, T. H., Enobong, I. B., & Atang, D. E. (2013). Comparative study of effect of honey on blood pressure and heart rate in healthy male and female subjects. British Journal of Medicine and Medical Research, 3(4), 2214.

Najafi, M., Zahednezhad, F., Samadzadeh, M., & Vaez, H. (2012). Zero flow global ischemia-induced injuries in rat heart are attenuated by natural honey. Advanced Pharmaceutical Bulletin, 2(2), 165.

Ebrahimi, M., Dehghani, F., Farhadian, N., Karimi, M., & Golmohammadzadeh, S. (2017). Investigating the anti-apoptotic effect of sesame oil and honey in a novel nanostructure form for treatment of heart failure. Nanomedicine Journal, 4(4), 245-253.

Aslan, A., Beyaz, S., Gok, O., Can, M. I., Parlak, G., Ozercan, I. H., & Gundogdu, R. (2021). Royal jelly abrogates flouride-induced oxidative damage in rat heart tissue by activating of the nrf-2/NF-κB and bcl-2/bax pathway. Toxicology Mechanisms and Methods, 31(9), 644-654.

Yook, T. H., Yu, J. S., & Jung, H. S. (2008). Effects of sweet bee venom and bee venom on the heart rate variability. Journal of Pharmacopuncture, 11(1), 41-54.

Response: I have not added more information about honey, because the effects of honey and its key active components on CVDs have been extensively characterized and described elsewhere (Olas, 2020). However, I have added new information about other bee products (in the manuscript and table 1). For example,

Results of He et al. (2018) indicate the protective action of total flavonoids of propolis (25 and 50 mg/kg/day) on pathological cardiac hypertrophy and heart failure in mice. The protective mechanism of tested propolis included PI3-K (phosphoinositide 3-kinase)/AKT signailing pathway.

Wang et al. (2015) observed that total flavonoids of propolis have inhibitory action on apoptosis of myocardial cells of chronic heart failure stimulated by adriamycin in rats. Moreover, used propolis preparation may regulate Cx43 expression, especially the phosphorylation status.

Majiene et al. (2006) noted inhibitory effect of propolis water solution on mitochnondrial respiration.

Recently, results of Aslan et al. (2021) have also demonstrated that royal jelly has cardioprotective potential against heart tissue damage – induced by fluoride (by activating of the nrf-2/NF-κB and bcl-2/bax signaling pathway). This experiment included 42 rats. Royal jelly was given in one dose: 100 mg/kg five times a week for 8 weeks.

Yook et al. (2008) show that sweet bee venom and bee venom may modulate heart rate variability. Guimaraes et al. (2004) also observed the changes in mean arterial pressure and heart rate in rats inoculated with Africanized bee venom.

  1. P2: “This was carried out on healthy and young and people”. Should be revised.

Response: I have corrected. Now, it is „healthy people”.

  1. P4: Figure 2. I think it is a table, not a figure.

Response: I have not corrected figure 1, because it has figure of bee and honey.

  1. P8, Figure 3: This figure does not make any sense and should be removed.

Response: I have removed figure 3.

  1. Some types of honey (mad-honey) have adverse effects on the cardiovascular system. Brief information could be given in the article considering the articles given below.

SOHN, C. H., Kim, W., Ahn, S., OH, B. J., KIM, W. Y., & LIM, K. S. (2005). Three cases of mad-honey poisoning presenting with cardiovascular emergencies. Journal of the Korean Society of Emergency Medicine, 322-325.

Shrestha, T. M., Nepal, G., Shing, Y. K., & Shrestha, L. (2018). Cardiovascular, psychiatric, and neurological phenomena seen in mad honey disease: a clinical case report. Clinical Case Reports, 6(12), 2355.

Ko, Y. G., Kim, K. H., Kim, A. J., Shin, D. W., Park, J. S., Roh, J. Y., & Ahn, J. Y. (2006). Two Cases of Mad-Honey Poisoning with Cardiovascular Symptom. Journal of The Korean Society of Clinical Toxicology, 4(1), 78-81.

Setareh-Shenas, S., Kaplin, S., Bania, T. C., & Kornberg, R. (2019). A Rare Case of Mad Honey Disease: A Reversible Cause of Complete Heart Block. JACC: Case Reports, 1(4), 579-582.

Response: I have not added more information about honey, because the effects of honey and its key active components on CVDs have been extensively characterized and described elsewhere (Olas, 2020). However, I have added short information about mad honey: “Some authors (Sohn et al., 2005; Ko et al., 2006; Shrestha et al., 2018; Setareh-Shenas et al., 2019) describe that mad honey have adverse effects on cardiovascular system. For example, these effects include hypotension, bradycardia and myocardial infarction.” (chapter of Introduction).

Round 2

Reviewer 1 Report

 The manuscript has improved significantly. The only minor problem remains is in figure 4, “atherosclerosis” is misleading, it is better to change this word to “anti-atherogenic”.

Author Response

The manuscript has improved significantly. The only minor problem remains is in figure 4, “atherosclerosis” is misleading, it is better to change this word to “anti-atherogenic”.

Response: I have corrected. Now, it is “anti-atherogenic”

Reviewer 2 Report

Thank you.

Author Response

Thank you for your help.